# MLE-STAR: Machine Learning Engineering Agent via Search and Targeted Refinement

**Jaehyun Nam**[1 2 *], **Jinsung Yoon**[1], **Jiefeng Chen**[1]
**Jinwoo Shin**[2], **Sercan Ö. Arık**[1], **Tomas Pfister**[1]
[1]Google Cloud, [2]KAIST
jaehyun.nam@kaist.ac.kr, jinsungyoon@google.com

## Abstract

Agents based on large language models (LLMs) for machine learning engineering (MLE) can automatically implement ML models via code generation. However, existing approaches to build such agents often rely heavily on inherent LLM knowledge and employ coarse exploration strategies that modify the entire code structure at once. This limits their ability to select effective task-specific models and perform deep exploration within specific components, such as experimenting extensively with feature engineering options. To overcome these, we propose *MLE-STAR*, a novel approach to build MLE agents. MLE-STAR first leverages external knowledge by using a search engine to retrieve effective models from the web, forming an initial solution, then iteratively refines it by exploring various strategies targeting specific ML components. This exploration is guided by ablation studies analyzing the impact of individual code blocks. Furthermore, we introduce a novel ensembling method using an effective strategy suggested by MLE-STAR. Our experimental results show that MLE-STAR achieves medals in 64% of the Kaggle competitions on the MLE-bench, significantly outperforming the best alternative.[1]

## 1 Introduction

The proliferation of machine learning (ML) has driven high-performance applications across diverse real-world scenarios, from fundamental tasks like tabular classification [1, 2, 3] to complex ones such as image denoising [4]. Despite these advances, developing such models remains a labor-intensive process for data scientists, involving extensive iterative experimentation and data engineering [5, 6]. To streamline such intensive workflows, recent research has focused on employing large language models (LLMs) [7, 8, 9] as *machine learning engineering (MLE) agents* [10, 11, 12]. By harnessing the coding and reasoning capabilities inherent in LLMs [13, 14], these agents conceptualize ML tasks as code optimization problems. They then navigate the potential code solutions ultimately producing executable code (*e.g.*, a Python script) based on a provided task description and dataset (see Figure 1).

Despite their promise as pioneering efforts, current MLE agents face several obstacles that limit their effectiveness. First, due to their strong reliance on inherent LLM knowledge, they are often biased toward familiar and frequently used methods (*e.g.*, the scikit-learn library [15] for tabular data), neglecting potentially promising task-specific methods. Additionally, these agents [10, 12] typically employ an exploration strategy that modifies the entire code structure at once in each iteration. This often results in agents pivoting prematurely to other steps (*e.g.*, model selection or hyperparameter tuning) because they lack the ability to perform deep, iterative exploration within specific pipeline components, such as experimenting different feature engineering options extensively.

---

[*]This work was done while Jaehyun was a student researcher at Google Cloud.
[1]We release open-source codebase of MLE-STAR at https://github.com/google/adk-samples.

39th Conference on Neural Information Processing Systems (NeurIPS 2025).

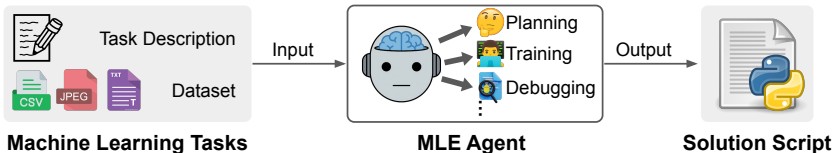

Figure 1: **Problem setup.** ML Engineering agents are designed to process a task description and datasets across various modalities (*e.g.*, tabular, text, image, audio, etc.) with the objective of determining the optimal solution for a given machine learning problem, such as classification, regression, sequence-to-sequence generation, image denoising, text normalization, etc.

**Contributions.** We propose **MLE-STAR**, a novel **ML E**ngineering agent that integrates web **S**earch and **TA**rgeted code block **R**efinement (see Figure 2 for an overview). Specifically, generating initial solution code, MLE-STAR utilizes Google Search to retrieve relevant and potentially state-of-the-art approaches that could be effective towards building a model. Moreover, to improve the solution, MLE-STAR extracts a specific code block that represents a distinct ML pipeline component, such as feature engineering or ensemble building, and then concentrates on exploring strategies that are targeted to that component, using previous attempts as feedback to reflect on. Here, to identify the code block that has the greatest impact on performance, MLE-STAR performs an ablation study that evaluates the contribution of each ML component. This refinement process is repeated, modifying various code blocks (*i.e.*, other ML components). In addition, we introduce a novel method to generate ensembles. MLE-STAR first proposes multiple candidate solutions. Then, instead of relying on a simple voting based on validation scores, MLE-STAR merges these candidates into a single improved solution using an ensemble strategy proposed by the agent itself. This ensemble strategy is iteratively refined based on the performance of the previous strategies.

To verify the effectiveness, we conduct comprehensive evaluations of MLE-STAR using the MLE-bench's Kaggle competitions [16]. The experimental results demonstrate that MLE-STAR, requiring only minimal human effort (*e.g.*, defining initial prompts that are generalizable to any tasks), significantly outperforms previous methods [12], including those requiring manual labor to collect strategies from Kaggle [10]. In particular, MLE-STAR achieves a substantial gain in medal achievement, improving it from 36.6% to 63.6% when compared to the top-performing baseline. Additionally, we show that our proposed ensemble technique provides a meaningful improvement to MLE-STAR.

## 2 Related work

**LLM agents.** Recent advances in LLMs have led to an active research in autonomous agents. General-purpose agents like ReAct [17] and HuggingGPT [18] typically use external tools to analyze various problems. Specialized agents, such as Voyager [19] for Minecraft or AlphaCode [20] for code generation, excel in specific domains, often using execution feedback to iteratively improve their approach. Extending these, we introduce MLE-STAR, an LLM agent that specialized in ML tasks.

**Automated machine learning.** Automated machine learning (AutoML) aims to reduce reliance on human experts by automating end-to-end ML pipelines [21, 22, 23]. Auto-WEKA [24], TPOT [25], and recent advances such as AutoGluon [26], have made progress through exploring within predefined model or hyperparameter spaces. AutoML research also specializes in areas such as neural network design [27, 28, 29, 30], and feature engineering [31, 32, 33, 34, 35]. However, these methods rely on predefined search spaces, which often require domain expertise to define. To address this, LLM-based MLE agents [10, 12], including MLE-STAR, are emerging, since they employ effective exploration strategies directly in the code space, without the need of manually-curated search spaces.

**MLE agents.** Leveraging coding and reasoning capabilities of LLMs [13, 14], research has been conducted on use of LLMs as MLE agents [11, 36, 37], which generate solution code, to automate ML workflows. While MLAB [38] and OpenHands [39] take general actions by calling tools to perform ML tasks, several studies specialize in ML automation. AIDE [12] generates candidate solutions in a tree structure to facilitate code space exploration. However, its heavy reliance on the LLM's internal knowledge can lead to outdated or overly simple model choices, and its refinement may prematurely shift focus between pipeline stages. DS-Agent [10] uses case-based reasoning [40, 41] to discover strategies for solution generation by utilizing manually curated cases (primarily from

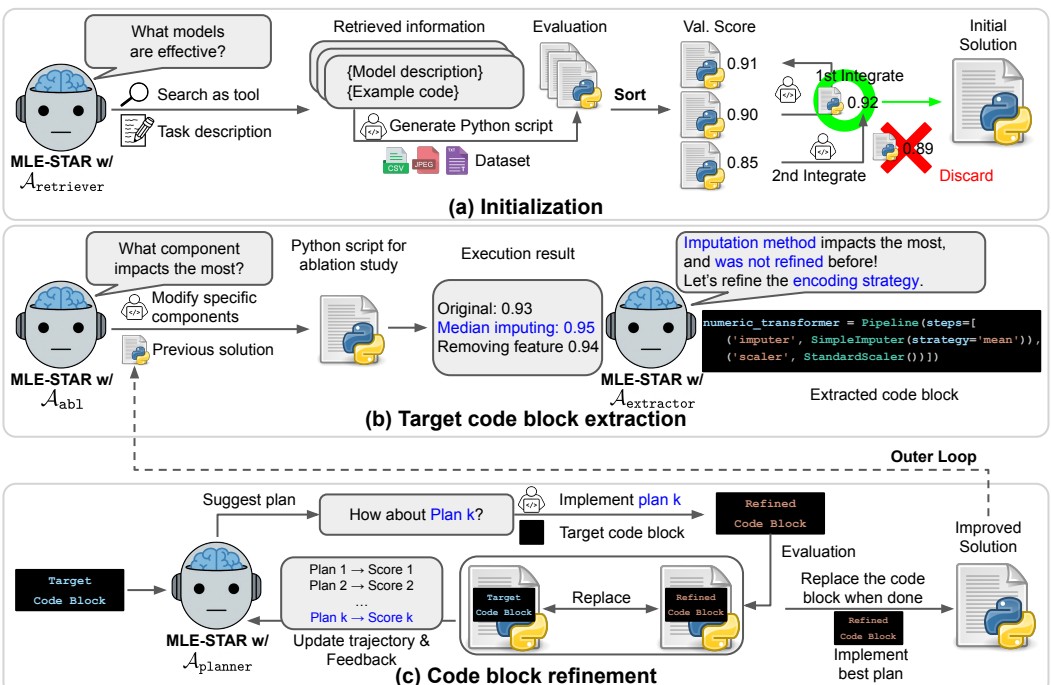

Figure 2: **Overview of MLE-STAR.** (a) Using search as a tool, MLE-STAR retrieves task-specific models and uses them to generate an initial solution. (b) In each refinement step, MLE-STAR performs an ablation study to extract the code block that have the greatest impact. Previously modified code blocks are also provided as feedback for diversity. (c) The extracted code block is iteratively refined based on plans suggested by the LLM, which explores various plans using previous experiments as feedback (*i.e.*, inner loop), and the target code block is also selected repeatedly (*i.e.*, outer loop, where the improved solution of (c) becomes the previous solution in (b)).

Kaggle). However, DS-Agent suffers from scalability issues due to its reliance on a manually built case bank, which requires significant human effort and can lead to solutions that are overfit to the source patterns. Also, it restricts applicability to novel task types (like complex multi-modal problems). Our method addresses these limitations. Instead of attempting to explore the broader code space or relying on a static case bank, MLE-STAR strategically explores implementation options for specific ML pipeline components. It also improves scalability by using LLMs with search as tool to retrieve effective models that fit the task beyond the constraints of a fixed case bank.

## 3 MLE-STAR

We introduce the proposed framework for MLE agents, MLE-STAR, that effectively leverages the coding and reasoning capabilities of LLMs to solve ML tasks. In a nutshell, our approach is based on first generating an initial solution by using web search as a tool (Section 3.1), and then refining solutions via nested loops. The outer loop targets one code block, which corresponds to the specific ML component extracted through an ablation study. The inner loop iteratively refines *only* this block until the outer loop moves to the next target (Section 3.2). We propose a novel ensemble method that improves the performance using the plan proposed by LLMs, which is iteratively refined (Section 3.3). To mitigate potential undesirable behaviors from LLMs, such as using test sample statistics for missing value imputation, we introduce specific modules (detailed in Section 3.4). The prompts and algorithms used in each step can be found in Appendix A and B, respectively.

**Problem setup.** Formally, our goal is to find an optimal solution $s^* = \arg\max_{s \in \mathcal{S}} h(s)$, where $\mathcal{S}$ is the space of possible solutions (*i.e.*, Python scripts) and $h : \mathcal{S} \to \mathbb{R}$ is a score function (*e.g.*, validation accuracy) [12]. To obtain $s^*$, we propose a multi-agent framework $\mathcal{A}$, which takes datasets $\mathcal{D}$ (that might contain multiple files) and a task description $\mathcal{T}_{\texttt{task}}$ (which includes task types, data

modalities, score functions, etc.) as input.[2] Here, $\mathcal{A}$ consists of $n$ LLM agents $(\mathcal{A}_1, \cdots, \mathcal{A}_n)$. Each agent $\mathcal{A}_i$ possesses specific functionalities, which are elaborated upon in following sections.

## 3.1 Generating an initial solution using web search as a tool

**Candidate model search.** MLE-STAR starts by generating an initial solution. For high performance in ML tasks, selecting the appropriate model is paramount. However, relying solely on an LLM for model suggestions can lead to suboptimal choices. For instance, we observe that LLMs propose models like logistic regression [15] even for competitions like jigsaw-toxic-comment-classification, which is a text classification task, potentially because LLMs favor familiar patterns from their pre-training data over up-to-date information. To mitigate this, we propose using web search as a tool for MLE-STAR first to retrieve $M$ effective, state-of-the-art models for the given task. This retrieved context is then used to guide the LLM in generating a more informed initial solution. Formally:

$$\{\mathcal{T}_{\texttt{model}}^i, \mathcal{T}_{\texttt{code}}^i\}_{i=1}^M = \mathcal{A}_{\texttt{retriever}}(\mathcal{T}_{\texttt{task}}), \tag{1}$$

where $\mathcal{T}_{\texttt{model}}$ represents the description of a retrieved model, while $\mathcal{T}_{\texttt{code}}$ provides corresponding example code. This example code is needed since the LLM can be unfamiliar with the model and cannot generate the executable code without proper guidance. Then, MLE-STAR involves evaluating of the performance of model $i$. To achieve this, candidate evaluation agent $\mathcal{A}_{\texttt{init}}$ first generates code, $s_{\texttt{init}}^i$, using the retrieved model to solve the given ML task. This process is formally defined as:

$$s_{\texttt{init}}^i = \mathcal{A}_{\texttt{init}}(\mathcal{T}_{\texttt{task}}, \mathcal{T}_{\texttt{model}}^i, \mathcal{T}_{\texttt{code}}^i). \tag{2}$$

We evaluate the performance of each $s$ using a task-specific metric $h$ on dataset $\mathcal{D}$. We denote the resulting score by $h(s)$, which encapsulates the entire process done in $s$: splitting $\mathcal{D}$ into training and validation sets, training the model specified in $s$ using the training data, and calculating $h$ on the validation data. The performance for $s_{\texttt{init}}^i$ is thus $h(s_{\texttt{init}}^i)$. As a result, a set of code scripts $\mathcal{S}_{\texttt{init}} = \{s_{\texttt{init}}^1, \cdots, s_{\texttt{init}}^M\}$ and their performance scores $\{h(s_{\texttt{init}}^1), \cdots, h(s_{\texttt{init}}^M)\}$ are obtained.

**Merging candidate models for initial solution.** After the evaluation of the $M$ retrieved models, a consolidated initial solution $s_0$ is constructed through an iterative merging procedure. Specifically, we first define $\pi$ be a permutation of the indices such that the scores are sorted in descending order: $h(s_{\texttt{init}}^{\pi(1)}) \geq h(s_{\texttt{init}}^{\pi(2)}) \geq \cdots \geq h(s_{\texttt{init}}^{\pi(M)})$. Then, we initialize the initial solution $s_0$ with the top-performing script, and record the current best score, *i.e.*, $s_0 \leftarrow s_{(1)}$, $h_{\texttt{best}} \leftarrow h(s_0)$, where $s_{(k)}$ denote the script $s_{\texttt{init}}^{\pi(k)}$ for simplicity. Finally, we sequentially attempt to incorporate the remaining scripts $s_{(k)}$ for $k = 2, \cdots, M$ into $s_0$. For each $k$, MLE-STAR creates a candidate merged script by leveraging an agent $\mathcal{A}_{\texttt{merger}}$ that attempts to integrate $s_{(k)}$ into the current $s_0$. Formally,

$$s_0 \leftarrow \mathcal{A}_{\texttt{merger}}(s_0, s_{(k)}), \quad h_{\texttt{best}} \leftarrow h(s_0) \tag{3}$$

where, $\mathcal{A}_{\texttt{merger}}$ is guided to introduce a simple average ensemble to merge multiple models. Finally, we merge the models until the validation score $h_{\texttt{best}}$ no longer improves (see Appendix B).

## 3.2 Refining a code block for solution improvement

The iterative refinement phase begins with an initial solution $s_0$ and proceeds for a predetermined number of $T$ outer loop steps, indexed by $t = 0, 1, \cdots, T-1$. At each step $t$, the goal is to improve the current solution $s_t$ to obtain $s_{t+1}$, optimizing for a performance metric $h$. This process involves two main stages: targeted code block extraction and code block refinement.

**Targeted code block extraction.** To effectively explore specialized improvement strategies, MLE-STAR identifies and targets specific code blocks within the ML pipeline represented by $s_t$. This selection is guided by an ablation study performed by an agent $\mathcal{A}_{\texttt{abl}}$. Specifically, the agent $\mathcal{A}_{\texttt{abl}}$ generates a code $a_t$ designed to perform an ablation study on $s_t$. This script creates variations of $s_t$ by modifying or disabling specific components. To encourage exploration of different pipeline parts across iterations, $\mathcal{A}_{\texttt{abl}}$ receives the summaries of previous ablation studies $\{\mathcal{T}_{\texttt{abl}}^i\}_{i=0}^{t-1}$ as input:

$$a_t = \mathcal{A}_{\texttt{abl}}(s_t, \{\mathcal{T}_{\texttt{abl}}^i\}_{i=0}^{t-1}). \tag{4}$$

---

[2]MLE-STAR works across any data modalities (*e.g.*, tabular, image, text, audio) and task types (*e.g.*, classification, image-to-image, sequence-to-sequence) – it is not restricted to specific inputs or objectives.

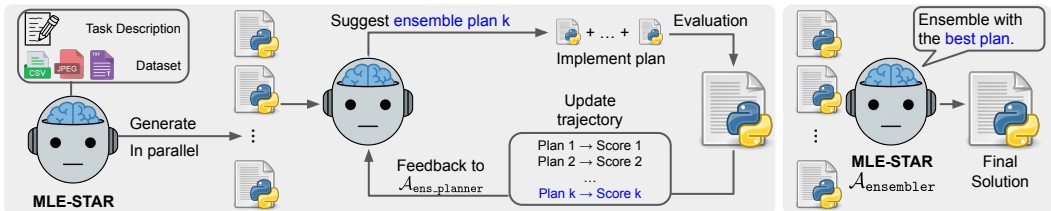

Figure 3: **Ensembling solutions.** MLE-STAR iteratively proposes effective ensemble strategies based on previous attempts, integrating multiple solutions generated in parallel into a single solution.

Then, $a_t$ is executed, producing output results $r_t$ (see examples in Appendix C). A summarization module $\mathcal{A}_{\texttt{summarize}}$ processes the script and its results to generate a concise ablation summary $\mathcal{T}_{\texttt{abl}}^t$:

$$\mathcal{T}_{\texttt{abl}}^t = \mathcal{A}_{\texttt{summarize}}(a_t, r_t). \tag{5}$$

Finally, MLE-STAR selects a critical block which has the most significant impact on performance. To achieve this, an extractor module $\mathcal{A}_{\texttt{extractor}}$ analyzes the $\mathcal{T}_{\texttt{abl}}^t$ to identify the code block $c_t$ within $s_t$ whose modification had the most significant impact on performance. To prioritize refinement of blocks not previously targeted, the set of already refined blocks $\{c_i\}_{i=0}^{t-1}$ is provided as context:

$$c_t, p_0 = \mathcal{A}_{\texttt{extractor}}(\mathcal{T}_{\texttt{abl}}^t, s_t, \{c_i\}_{i=0}^{t-1}). \tag{6}$$

Here, MLE-STAR also generates the initial plan $p_0$ for code block refinement at the same time, since $\mathcal{T}_{\texttt{abl}}$ can provide a good starting point by analyzing the modification of corresponding component.

**Code block refinement.** Once the targeted code block $c_t$ is defined, MLE-STAR explores various refinement strategies to improve the metric $h$. This involves an inner loop exploring $K$ potential refinement for $c_t$. An agent $\mathcal{A}_{\texttt{coder}}$ first implements $p_0$, transforming $c_t$ into a refined block $c_t^0$, *i.e.*, $c_t^0 = \mathcal{A}_{\texttt{coder}}(c_t, p_0)$. A candidate solution $s_t^0$ is formed by substituting $c_t^0$ into $s_t$:

$$s_t^0 = s^t.\texttt{replace}(c_t, c_t^0), \tag{7}$$

where, $\texttt{replace}$ denotes the code replacement operation. Finally, the performance $h(s_t^0)$ is evaluated.

To discover potentially more effective or novel refinement strategies, MLE-STAR iteratively generates and evaluates further plans. For $k = 1, \cdots, K - 1$, a planning agent $\mathcal{A}_{\texttt{planner}}$ proposes the next plan $p_k$. This agent leverages the previous attempts within the current outer step $t$ as feedback:

$$p_k = \mathcal{A}_{\texttt{planner}}(c_t, \{(p_j, h(s_t^j))\}_{j=0}^{k-1}). \tag{8}$$

For each plan $p_k$, the coding agent generates the corresponding refined block, *i.e.*, $c_t^k = \mathcal{A}_{\texttt{coder}}(c_t, p_k)$, creates the candidate solution $s_t^k = s_t.\texttt{replace}(c_t, c_t^k)$, and evaluates its performance $h(s_t^k)$. After exploring $K$ refinement strategies (indexed $k = 0, \cdots, K - 1$), the best-performing candidate solution is identified: $k^* = \arg\max_{k \in \{0, \cdots, K-1\}} h(s_t^k)$. The solution for the next outer step, $s_{t+1}$, is updated to $s_t^{k^*}$ only if an improvement over $s_t$ is found. This iterative process continues until $t = T$.

### 3.3 Further improvement by exploring ensemble strategies

To further improve upon the best single solution generated, we introduce a novel ensembling procedure (Figure 3). Standard practice might involve generating multiple candidate solutions and selecting the one with the highest score [42] according to metric $h$. However, analogous to model ensembling, we posit that suboptimal solutions might contain complementary strengths, and combining multiple solutions could lead to superior performance compared to relying on any single one. Therefore, we employ the planning capabilities of MLE-STAR to automatically discover effective strategies for ensembling. Specifically, let $\{s_l\}_{l=1}^L$ be a set of $L$ distinct solutions obtained (*e.g.*, from parallel runs of the process described earlier). Our goal is to find an effective ensemble plan $e$ that merges these solutions, which mirrors the structure of the targeted code block refinement stage. We start with an initial ensemble plan $e_0$ (*e.g.*, a simple strategy like averaging the final predictions obtained from the models trained using each solution $s_l$), proposed by MLE-STAR itself. After the performance $h(s_{\texttt{ens}}^0)$ for the initial plan $e_0$ is calculated, for a fixed number of iterations, $r = 1, \cdots, R$, the planning agent $\mathcal{A}_{\texttt{ens\_planner}}$, specialized in suggesting ensemble plans, proposes subsequent ensemble plans $e_r$.

This agent uses the history of previously attempted ensemble plans and their resulting performance as feedback, *i.e.*, $e_r = \mathcal{A}_{\texttt{ens\_planner}}(\{s_l\}_{l=1}^L, \{(e_j, h(s_{\texttt{ens}}^j))\}_{j=0}^{r-1})$. Each $e_r$ is implemented via $\mathcal{A}_{\texttt{ensembler}}$ to obtain $s_{\texttt{ens}}^r$:

$$s_{\texttt{ens}}^r = \mathcal{A}_{\texttt{ensembler}}(e_r, \{s_l\}_{l=1}^L). \tag{9}$$

Finally, after exploring $R$ ensemble strategies, the ensemble result that achieves the highest performance is selected as the final output, yielding the final ensembled result $s_{\texttt{ens}}^* = s_{\texttt{ens}}^{r^*}$: $r^* = \arg\max_{r \in \{0, \dots, R\}} h(s_{\texttt{ens}}^r)$. This procedure allows MLE-STAR to autonomously explore and identify potentially novel and effective ways to combine multiple complex solutions.

### 3.4 Additional modules for robust MLE agents

**Debugging agent.** We detail the design of our debugging agent within MLE-STAR. If the execution of a Python script $s$ triggers an error, resulting in a record $\mathcal{T}_{\texttt{bug}}$ (*e.g.*, a traceback), MLE-STAR employs a debugging module $\mathcal{A}_{\texttt{debugger}}$ to attempt correction. This process iteratively updates the script:

$$s \leftarrow \mathcal{A}_{\texttt{debugger}}(s, \mathcal{T}_{\texttt{bug}}). \tag{10}$$

The debugging step is repeated until either the script executes successfully, or a predefined maximum number of debugging rounds is reached. If the bug cannot be resolved, MLE-STAR proceeds to the next task using the latest version of the script that is known to be executable.

**Data leakage checker.** We observe that LLM-generated Python scripts might have the risk of introducing data leakage, for example, by improperly accessing information from a test dataset during training dataset preparation (see Figure 6). To address this, we introduce a checker agent, $\mathcal{A}_{\texttt{leakage}}$, which analyzes the solution script $s$ prior to its execution. Recognizing that full-script analysis can be inefficient for lengthy code, we adopt a targeted approach. First, we extract the code block $c_{\texttt{data}}$ where data preprocessing is done. Second, $c_{\texttt{data}}$ is passed to the checker. If $\mathcal{A}_{\texttt{leakage}}$ detects potential data leakage, it generates a corrected version $c_{\texttt{data}}^*$: $c_{\texttt{data}}^* = \mathcal{A}_{\texttt{leakage}}(c_{\texttt{data}})$. Finally, the original script $s$ is updated by replacing the identified segment with its corrected version: $s \leftarrow s.\texttt{replace}(c_{\texttt{data}}, c_{\texttt{data}}^*)$. If no leakage is detected in $c_{\texttt{data}}$ by $\mathcal{A}_{\texttt{leakage}}$, the script $s$ remains unmodified. All generated solutions are passed through a data leakage checker, $\mathcal{A}_{\texttt{leakage}}$, prior to their execution for evaluation.

**Data usage checker.** We observe that LLM-generated scripts sometimes neglect using provided data sources, focusing solely on simple formats like CSVs (see Figure 7). To ensure the utilization of all relevant provided data, MLE-STAR introduces a data usage checker agent, $\mathcal{A}_{\texttt{data}}$. Specifically, before MLE-STAR starts refinement, $\mathcal{A}_{\texttt{data}}$ checks the initial solution $s_0$ along with the task description $\mathcal{T}_{\texttt{task}}$. If relevant provided data is not adequately used, $\mathcal{A}_{\texttt{data}}$ revises the initial script as:

$$s_0 \leftarrow \mathcal{A}_{\texttt{data}}(s_0, \mathcal{T}_{\texttt{task}}). \tag{11}$$

## 4 Experiments

In this section, we validate the effectiveness of MLE-STAR using 22 Kaggle competitions from MLE-bench Lite [16]. Our results demonstrate that MLE-STAR significantly outperforms baselines, including those employing various LLMs (Section 4.1). Furthermore, we show that using better models and leveraging our proposed ensemble strategy effectively improves performance (Section 4.2). We also provide the example solutions generated by MLE-STAR, in Appendix D.

**Common setup.** All experiments are conducted on 22 Kaggle competitions from MLE-bench Lite [16] using three random seeds, unless otherwise specified. Here, we use an agent $\mathcal{A}_{\texttt{test}}$, which takes the task description and the final solution as input, and outputs the code that incorporates loading test sample and creating a submission file (see Appendix E for details). MLE-STAR begins by retrieving four model candidates. MLE-STAR refines for four inner loops, while exploring four outer loops. For ensemble, MLE-STAR generates two solutions in parallel, and explore ensemble strategies for five rounds. Following the MLE-bench's setup, we set a maximum time limit of 24 hours for a fair comparison (see computation analysis in Appendix F). We primarily consider AIDE [12] as our main baseline, given its state-of-the-art performance on MLE-bench. It is important to note that other baselines often limit their generalizability across various task types (*e.g.*, audio classification, sequence-to-sequence), frequently showcasing results only on simpler modalities like tabular [11, 37]. For instance, DS-Agent [10] requires a manually constructed case bank, and their current GitHub repository lacks cases for audio classification, sequence-to-sequence, image classification, etc.

Table 1: **Main results from MLE-bench Lite.** Each experiment is repeated using three seeds, except for o1-preview (AIDE) and GPT-4o (AIDE), which use 16 and 36 seeds, respectively. All results are taken from the GitHub repository of MLE-bench paper [16], except for the model using Gemini-2.0-Flash and Gemini-2.5-Pro. Scores represent the mean and one standard error of the mean.

| Model | Made Submission (%) | Valid Submission (%) | Above Median (%) | Bronze (%) | Silver (%) | Gold (%) | Any Medal (%) |
|---|---|---|---|---|---|---|---|
| **MLE-STAR (Ours)** | | | | | | | |
| **gemini-2.5-pro** | **100.0**±0.0 | **100.0**±0.0 | **83.3**±4.6 | 6.1±3.0 | **21.2**±5.1 | **36.4**±6.0 | **63.6**±6.0 |
| gemini-2.0-flash | 95.5±2.6 | 95.5±2.6 | 63.6±6.0 | **9.1**±3.6 | 4.5±2.6 | 30.3±5.7 | 43.9±6.2 |
| **AIDE [12]** | | | | | | | |
| gemini-2.0-flash | 87.9±4.0 | 78.8±5.0 | 39.4±6.0 | 4.5±2.6 | 9.1±3.5 | 12.1±4.0 | 25.8±5.4 |
| o1-preview | 99.7±0.3 | 90.3±1.6 | 58.2±2.6 | 4.8±1.1 | 11.1±1.7 | 20.7±2.2 | 36.6±2.6 |
| gpt-4o | 82.1±1.4 | 65.7±1.7 | 29.9±1.6 | 3.4±0.6 | 5.8±0.8 | 9.3±1.0 | 18.6±1.4 |
| llama-3.1-405b-instruct | 72.7±5.5 | 51.5±6.2 | 18.2±4.7 | 0.0±0.0 | 4.5±2.6 | 6.1±2.9 | 10.6±3.8 |
| claude-3-5-sonnet | 81.8±4.7 | 66.7±5.8 | 33.3±5.8 | 3.0±2.1 | 6.1±2.9 | 10.6±3.8 | 19.7±4.9 |
| **MLAB [38]** | | | | | | | |
| gpt-4o | 84.8±4.4 | 63.6±5.9 | 7.6±3.3 | 3.0±2.1 | 1.5±1.5 | 1.5±1.5 | 6.1±2.9 |
| **OpenHands [39]** | | | | | | | |
| gpt-4o | 81.8±4.7 | 71.2±5.6 | 16.7±4.6 | 3.0±2.1 | 3.0±2.1 | 6.1±2.9 | 12.1±4.0 |

Table 2: Comparison with DS-Agent.

| Task | Metric | DS-Agent | **MLE-STAR** |
|---|---|---|---|
| WBY | MAE (↓) | 213 | **166** |
| MCC | RMLSE (↓) | 0.2964 | **0.2911** |
| ST | Accuracy (↑) | 0.7982 | **0.8091** |
| ES | AUROC (↑) | 0.8727 | **0.9101** |

Table 3: Performance with Claude-Sonnet-4.

| Task | Metric | 2.0-Flash | **Sonnet-4** |
|---|---|---|---|
| DDD | RMSE (↓) | 0.0681 | **0.0155** |
| DBI | Log Loss (↓) | 0.4535 | **0.3114** |
| SAI | Log Loss (↓) | 0.2797 | **0.2610** |
| WCR | AUROC (↑) | **0.9903** | 0.9888 |

## 4.1 Main results

**Quantitative results.** As demonstrated in Table 1, MLE-STAR significantly enhances the performance of various baseline models. For instance, when applied to Gemini-2.0-Flash, MLE-STAR improves AIDE's any medal achieving rates in Kaggle competitions from 25.8% to 43.9%, representing an improvement of over 18 percentage points, and rate of above median from 39.4% to 63.6%. Notably, MLE-STAR with Gemini-2.0-Flash also substantially outperforms AIDE using a powerful reasoning model (*i.e.*, o1-preview) in terms of achieving gold medals in 10% more tasks. Moreover, using Gemini-2.5-Pro, MLE-STAR shows a medal achievement of over 60%.

**Comparison to DS-Agent.** While DS-Agent [10] shows competitive results on ML tasks, it necessitates human effort to curate its case bank from Kaggle. Consequently, a direct comparison between DS-Agent and AIDE or our method is not feasible, as collecting tasks across diverse modalities, such as audio classification or image denoising, requires additional effort. Nevertheless, we utilize four tabular classification tasks, *i.e.*, wild-blueberry-yield (WBY), media-campaign-cost (MCC), spaceship-titanic (ST), and enzyme-substrate (ES), the same ones employed during DS-Agent's development stage [10], for a comparison. All experiments are done for 5 seeds following the original setup. As shown in Table 2, MLE-STAR significantly outperforms DS-Agent even without human efforts. See Appendix G for additional results, including comparison with AutoGluon [26].

## 4.2 Ablation studies

**Performance with an advanced reasoning model.** To assess if a more advanced reasoning model could enhance MLE-STAR's performance, we conduct an experiment with the recently released advanced reasoning models. First of all, as shown in Table 1, Gemini-2.5-Pro [43] yields better performance than using Gemini-2.0-Flash. For example, in denoising-dirty-documents competition, MLE-STAR with Gemini-2.0-Flash scored above the median across all three seeds, failing to achieve any medals. However, when using Gemini-2.5-Pro, MLE-STAR achieves two gold medals and one silver medal. These results demonstrate that MLE-STAR is designed to harness the advancements of rapidly improving reasoning-based LLMs.

Table 4: **Ablation on ensemble strategy.** Experiment results on MLE-bench Lite, repeated three seeds using Gemini-2.0-Flash. Scores represent the mean and one standard error of the mean.

| Ensemble strategy | Made Submission (%) | Valid Submission (%) | Above Median (%) | Bronze (%) | Silver (%) | Gold (%) | Any Medal (%) |
|---|---|---|---|---|---|---|---|
| **AIDE [12]** | | | | | | | |
| None | $87.9_{\pm4.0}$ | $78.8_{\pm5.0}$ | $39.4_{\pm6.0}$ | $4.5_{\pm2.6}$ | $9.1_{\pm3.5}$ | $12.1_{\pm4.0}$ | $25.8_{\pm5.4}$ |
| **MLE-STAR (Ours)** | | | | | | | |
| None | $\mathbf{95.5}_{\pm2.6}$ | $\mathbf{95.5}_{\pm2.6}$ | $57.6_{\pm6.1}$ | $7.6_{\pm3.3}$ | $4.5_{\pm2.6}$ | $25.8_{\pm5.4}$ | $37.9_{\pm6.0}$ |
| Best-of-N | $\mathbf{95.5}_{\pm2.6}$ | $\mathbf{95.5}_{\pm2.6}$ | $62.1_{\pm6.0}$ | $6.1_{\pm3.0}$ | $7.6_{\pm3.3}$ | $28.8_{\pm5.6}$ | $42.4_{\pm6.1}$ |
| Average ensemble | $\mathbf{95.5}_{\pm2.6}$ | $\mathbf{95.5}_{\pm2.6}$ | $60.6_{\pm6.1}$ | $6.1_{\pm3.0}$ | $\mathbf{12.1}_{\pm4.0}$ | $25.8_{\pm9.4}$ | $\mathbf{43.9}_{\pm6.2}$ |
| **Ours** | $\mathbf{95.5}_{\pm2.6}$ | $\mathbf{95.5}_{\pm2.6}$ | $\mathbf{63.6}_{\pm6.0}$ | $\mathbf{9.1}_{\pm3.6}$ | $4.5_{\pm2.6}$ | $\mathbf{30.3}_{\pm5.7}$ | $\mathbf{43.9}_{\pm6.2}$ |

Table 5: **Sensitivity analysis on the number of ensemble rounds.** Experiment results on 4 tasks from MLE-bench Lite, repeated three seeds using Gemini-2.0-Flash. We report the mean score.

| Ensemble Round | DDD (RMSE; ↓) | DBI (Log Loss; ↓) | SAI (Log Loss; ↓) | WCR (AUROC; ↑) |
|---|---|---|---|---|
| 1 | 0.07147 | **0.45351** | 0.28164 | 0.98943 |
| 3 | **0.06805** | **0.45351** | **0.27967** | 0.98898 |
| 5 | **0.06805** | **0.45351** | **0.27967** | **0.99028** |

Table 6: **Ablation on proposed components.** Experiment results on 4 tasks from MLE-bench Lite, repeated three seeds using Gemini-2.0-Flash. We report the mean score and bold the best one.

| Targeted Refinement | Search Tool | DDD (RMSE; ↓) | DBI (Log Loss; ↓) | SAI (Log Loss; ↓) | WCR (AUROC; ↑) |
|---|---|---|---|---|---|
| ✗ | ✓ | 0.10818 | 0.45689 | 0.29141 | 0.98532 |
| ✓ | ✗ | 0.09303 | 0.65242 | 0.30529 | 0.96509 |
| ✓ | ✓ | **0.06805** | **0.45351** | **0.27967** | **0.99028** |

In addition, we conduct additional experiments using Claude-Sonnet-4. Here, we select four different type of competitions: image-to-image (denoising-dirty-documents; DDD), image classification (dog-breed-identification; DBI), text classification (spooky-author-identification, SAI), and audio classification (the-icml-2013-whale-challenge-right-whale-redux; WCR). We run each competition for three seeds. As shown in Table 3, Claude-Sonnet-4 also shows promising results, indicating that our framework is also compatible and generalizable in terms of LLM type.

**Effectiveness of proposed ensemble method.** As highlighted in Table 4, MLE-STAR demonstrates a significant performance improvement over the competing baseline, *i.e.*, AIDE, achieving over a 12% higher rate of obtaining any medal *even without* additional ensemble strategy. Notably, by ensembling multiple solution candidates, our approach yields even greater performance gains, *i.e.*, MLE-STAR consistently improves the success rate for achieving any medal (and specifically gold medals), also surpassing the median human expert's performance by a larger margin compared to scenarios where this ensembling method is not used. While simpler strategies, such as selecting the solution with the best validation score or averaging final submissions, also offer benefits, MLE-STAR shows stronger effectiveness, *e.g.*, leading to a higher number of gold medals.

Furthermore, we conduct a sensitivity analysis on the number of ensemble rounds. Here, we utilize four datasets as same as Table 3. Table 5 indicates that while we utilize five rounds for ensemble strategy exploration, comparable performance can be achieved with fewer rounds.

**Effectiveness of proposed components.** Here, we focus on two key components of the proposed approach: *targeted refinement strategy* and *integrating search as a tool*. As shown in Table 6, we verify the effectiveness of our targeted refinement and the benefits of the search tool integration. Specifically, our findings indicate that refining only the specific code block identified by the ablation study agent is more effective than refining the entire codebase. Moreover, our experiments consistently show that utilizing search as a tool significantly enhances the overall performance.

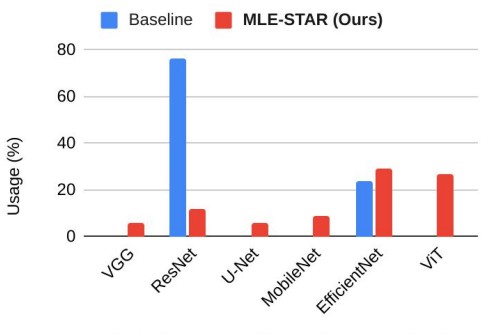

Figure 4: **Model usage** (%) on image classification competitions. Other models (11.7%), which are used by MLE-STAR, are omitted.

Figure 5: **Human intervention.** By manually adding a model description, MLE-STAR integrates its training into the framework.

```
Improperly imputed missing values
# Combined DataFrame for Consistent Preprocessing
all_df = pd.concat([train_df, test_df])
# Impute Numerical features with median
num_cols = ['Age', 'RoomService', 'FoodCourt']
for col in num_cols:
  if all_df[col].isnull().any():
    median_val = all_df[col].median()
    all_df[col] = all_df[col].fillna(median_val)
```

```
Python script after refined by data leakage checker
# Preprocess Training Set and get fit statistics
X_train_processed, stats =
    preprocess_data(X_train, fit_stats=None)
# Preprocess Test Set using statistics from Train Set
X_test_processed =
    preprocess_data(X_test, fit_stats=stats)
```

Figure 6: MLE-STAR's **data leakage checker** introduces appropriate preprocessing.

```
Python script before refined by data usage checker
data = pd.read_csv('./input/train.csv')
```

```
Python script after refined by data leakage checker
def process_xyz_files(df): # Features using XYZ files
  def volume(filepath):
    lines = open(filepath, 'r').readlines()
    a_vec = list(map(float, lines[2].split()[1:]))
    b_vec = list(map(float, lines[3].split()[1:]))
    c_vec = list(map(float, lines[4].split()[1:]))
    volume = np.dot(a_vec, np.cross(b_vec, c_vec))
    return abs(volume)
  for index, row in df.iterrows():
    file_path = os.path.join(row['id'],'geometry.xyz')
    df.loc[index, 'atomic_volume'] = volume(file_path)
  return df
data = process_xyz_files(data) # Process the train data
```

Figure 7: MLE-STAR's **data usage checker** captures previously unused information.

## 5   Discussion

**Qualitative observations on selected models.** Figure 4 illustrates the model usage of two MLE agents: AIDE and MLE-STAR. AIDE primarily employs ResNet [44] for image classification. However, ResNet, released in 2015, is now considered outdated and can result in suboptimal performance. In contrast, our MLE-STAR primarily utilizes more recent and competitive models like Efficient-Net [45] or ViT [46], leading to the performance gain, winning 37% of the medals, more than AIDE, which wins 26% of the image classification challenges.

**Human intervention.** MLE-STAR readily adopts even more recent models with minimal human intervention. While MLE-STAR automatically constructs a model description $\{\mathcal{T}_{\texttt{model}}, \mathcal{T}_{\texttt{code}}\}$ using search as tool, a natural extension involves leveraging human expertise for this construction. As shown in Figure 5, by manually adding a model description for RealMLP [47], MLE-STAR successfully integrates its training into the framework, a model not previously retrieved. In addition, users can also specify the target code blocks by replacing the ablation summary with manually written instructions.

**Misbehavior of LLMs and corrections.** We observe that while the code generated by the LLM executed correctly, their content is sometime unrealistic, exhibiting hallucination. For example, Figure 6 illustrates an impractical approach where test data is preprocessed using its own statistics. Since test data must remain unseen, correction in the code is necessitated, for which, MLE-STAR employs a data leakage checker $\mathcal{A}_{\texttt{leakage}}$ to identify such issues in the generated Python script. If a problem is detected, MLE-STAR refines the code. As shown in the Figure, MLE-STAR successfully identifies the issue and modifies the code by, first extracting statistics from the training data and then preprocessing the test data using these calculated statistics. In addition, the improvement process

Table 7: Improvement failure when not using data leakage checker $\mathcal{A}_{\texttt{leakage}}$ on spaceship-titanic competition.

| Metric | Accuracy ($\uparrow$) |
|---|---|
| Validation | $0.8188 \rightarrow \mathbf{0.8677}$ |
| Test | $\mathbf{0.8033} \rightarrow 0.7343$ |

Table 8: Ablation study of data usage checker $\mathcal{A}_{\texttt{data}}$ on nomad2018-predicting competition.

| Model | $\mathcal{A}_{\texttt{data}}$ | RMSLE ($\downarrow$) |
|---|---|---|
| MLE-STAR | ✗ | 0.0591 |
| **MLE-STAR** | ✓ | **0.0559** |

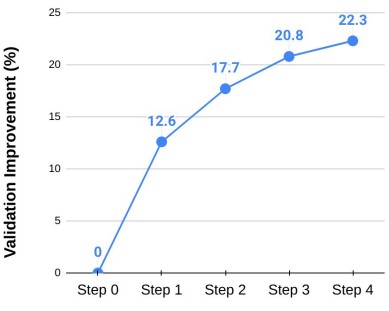

Figure 8: Solution refinement trajectory.

can fail to generalize when $\mathcal{A}_{\texttt{leakage}}$ is not employed, as exemplified in Table 7. In this example, the validation accuracy (*i.e.*, the target objective) improves, but the test accuracy drops significantly. This is attributed to the LLM performing feature engineering using the target variable `Transported`, which is not accessible in the test set, leading to data leakage and subsequently, poor test performance.

We also observe that LLMs often generate Python scripts that overlook some of the provided data sources. For example, in the nomad2018-predicting competition, Gemini-2.0-Flash solely loads train.csv, neglecting the use of geometry.xyz (see Figure 7). To address this, MLE-STAR employs $\mathcal{A}_{\texttt{data}}$, which reexamines the task description to ensure that all given data is utilized. As shown in Figure 7, this design enables MLE-STAR to incorporate previously neglected data. As a result, performance is significantly improved, as shown in Table 8.

**Progressive improvement via MLE-STAR refinement.** This section details the progressive improvement of solutions achieved by MLE-STAR, as measured by validation metrics. Given the task-specific nature of evaluation metrics, we report the average relative error reduction (%) across the all 22 challenges in MLE-bench Lite [16]. This metric measures the extent to which MLE-STAR reduces the error of an initial solution. Figure 8 demonstrates a consistent improvement as MLE-STAR proceeds through its refinement steps, which each step focusing on refining a single code block via an inner loop. Significantly, the magnitude of improvement is notable in the early refinement stages. We posit that this stems from MLE-STAR's ablation study module which helps to target the most influential code blocks for modification first.

**Discussion on potential plagiarism.** Following MLE-bench, we utilize Dolos [48], a source code plagiarism detection tool, to analyze generated solution code by MLE-STAR, against the top associated notebooks (*i.e.*, Jupyter notebook) from each Kaggle competition. Our analysis, summarized in Table 13 (see Appendix L), shows that no final solution code and notebook pair exceeded a 60% similarity score (*i.e.*, a criteria suggested from the MLE-bench paper), indicating no detected instances of plagiarism. This also shows that MLE-STAR's solution is sufficiently new compared to the existing solutions in Kaggle.

# 6 Conclusion

We propose MLE-STAR, a novel MLE agent designed for various ML tasks. Our key idea is to utilize a search engine to retrieve effective models and then explore various strategies targeting specific ML pipeline components to improve the solution. The effectiveness of MLE-STAR is validated by winning medals in 64% (where 36% are gold medals) of the MLE-bench Kaggle competitions.

**Limitation.** We acknowledge that MLE-STAR requires higher cost due to increased token usage. We include corresponding cost analysis in Appendix K. However, it is worth to note that still, with Gemini-2.0-Flash, the cost of MLE-STAR is only about $0.24 per each ML challenge. In addition, since Kaggle competitions are publicly accessible, there is a potential risk that LLMs might have been trained with the relevant discussions about the challenge. Nevertheless, we show that MLE-STAR's solution is sufficiently novel (using LLM as a judge) compared to the discussions on Kaggle (see Appendix H), and also show that its similarity compared to notebooks on Kaggle does not exceed 60%, alleviating such plagiarism issue (see Appendix L).

## Acknowledgements and disclosure of funding

We would like to thank Raj Sinha, Subin Kim, Changyeon Kim, Dongjun Lee, Jihoon Tack, and anonymous reviewers for their helpful feedback and discussions. This work was partly supported by Institute for Information & communications Technology Planning & Evaluation (IITP) grant funded by the Korea government (MSIT) (RS-2019-II190075, Artificial Intelligence Graduate School Program (KAIST)) and ITRC (Information Technology Research Center) grant funded by the Korea government (MSIT) (IITP-2025-RS-2024-00436857, 50%).

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
