# OpenReview forum: "MLE-STAR: Machine Learning Engineering Agent via Search and Targeted Refinement"
_NeurIPS.cc/2025/Conference — NeurIPS 2025 poster_

### Official Review · Reviewer_3VFF · 2025-06-26

**Clarity:** 3
**Significance:** 3
**Originality:** 3
**Rating:** 5
**Confidence:** 4

**Summary:**

This paper considers the problem of code generation for machine learning tasks using language models.
However, language models tend to use methods they are familiar with, possibly caused by the training data distribution.
So the paper proposes to do the following:

- Use web search to retrieve task-relevant model descriptions and code examples.

- Generate an initial solution script using the retrieved information. Evaluate the performance of all the scripts on some validation set. Merge all the scripts into a single solution.

- Perform ablation studies to identify the most impactful part of the pipeline and iteratively refines the solution.

- Combine multiple solutions using an LLM-driven ensemble strategy to produce the final output.

Empirically, the proposed AI agent outperforms the baseline methods (AIDE, and DS-Agent when comparison is possible) on MLE-bench Lite using different backbone LLMs.

**Questions:**

The agent identifies one block and refines it at a time. Are there cases where multi-block refinement or full model resets (e.g., changing the model entirely) would be more effective?

Does the paper consider metrics beyond model performance, such as code readability and modularity?

**Ethical Concerns:**

["NO or VERY MINOR ethics concerns only"]

**Final Justification:**

The authors have addressed all the concerns in the review. The additional ablation results have further justified the algorithm design. The authors have also agreed to add discussions to make the contributions clearer.

**Limitations:**

The limitations are discussed in Section 6.

**Quality:**

4

**Strengths And Weaknesses:**

### Strengths

This paper contributes an agentic system that simulates an MLE's task, involving selecting a model, running evaluations, doing ablation studies and improving the model with the results.
It shows solid results on the MLE-bench Lite dataset.

The paper is well written and notationally clear. The overview figures (Figure 2 and 3) are helpful to understand the pipeline.

### Weaknesses

**Need a more comprehensive ablation study**: The ablation studies have analyzed the effectiveness of using an advanced reasoning LLM (which unsurprisingly improves the performance) and the effectiveness of different ensemble methods.
However, we may need more analytical studies to understand the effectivness of other critical components.
For example, is web search necessary -- what is the performance if the agent generates a set of models by calling an LLM?
How accurate is the ablation agent; how often does it find a part that shows no improvement after refinement?

**Clarity on novelty**
There is prior work on writing an academic paper [1], which also involves web search and an agent system to perform trial-and-error.
It would be helpful to clarify the novelty of this work.

[1] Lu, Chris, et al. "The ai scientist: Towards fully automated open-ended scientific discovery." arXiv preprint arXiv:2408.06292 (2024).

---

> ### Author Rebuttal · Authors · 2025-07-30
>
> Dear Reviewer 3VFF,
>
> We deeply appreciate your invaluable time and efforts in reviewing our manuscript. We respond to each of your comments one-by-one in what follows.
>
> ---
> **[W1-1] What is the performance if the agent generates a set of models by calling an LLM?**
>
> Thanks for your constructive feedback and we conducted an additional ablation study, following your suggestion. Specifically, we removed the search tool and ran the experiments with the same configuration as the current manuscript. As shown in the table below, using search as a tool consistently enhances the MLE-STAR performance. We will incorporate this ablation study in the revised manuscript.
> \begin{array}{lccccc}
> \hline
> \text{Method}&\text{Search as a tool}&\text{DDD (RMSE)}&\text{DBI (Log Loss)}&\text{SAI (Log Loss)}&\text{WCR (AUROC)}\newline
> \hline
> \textbf{MLE-STAR (Variant)}&\text{X}&0.09303&0.65242&0.30529&0.96509\newline
> \textbf{MLE-STAR}&\text{O}&\textbf{0.06805}&\textbf{0.45351}&\textbf{0.27967}&\textbf{0.99028}\newline
> \hline
> \end{array}
>
> **Experimental settings:** We used the same four datasets – denoising-dirty-documents (DDD), dog-breed-identification (DBI), spooky-author-identification (SAI), and the-icml-2013-whale-challenge-right-whale-redux (WCR) – used in Table 3 of our manuscript. The experiments were repeated three times for each dataset and reported the average metrics (lower the better except WCR).
>
> ---
> **[W1-2] How accurate is the ablation agent; how often does it find a part that shows no improvement after refinement?**
>
> We would like to first clarify that Figure 8 (in Section 5 of the current manuscript) already demonstrates that the validation score progressively improves throughout the refinement steps, indicating our ablation study agent effectively targets the influential code blocks early on, as evidenced by the significant improvements in the initial refinement stages.
>
> To further analyze, we conducted additional analysis and found that 38.4% of refinement steps showed no improvement, with this occurring more frequently in later stages. This finding aligns with Figure 8, which shows smaller improvements as the outer loop progresses. We will incorporate this discussion in the revised manuscript.
>
> ---
> **[W2] Novelty: There is prior work on writing an academic paper [1], which also involves web search and an agent system to perform trial-and-error. It would be helpful to clarify the novelty of this work.**
>
> We appreciate the opportunity to clarify the novelty of our work compared to Lu et al. [1]. While Lu et al. also involve web search and agent systems for trial-and-error, their primary focus is on automating the end-to-end scientific research process, from idea generation to experimental execution. This differs significantly from our work in several key aspects.
>
> Unlike Lu et al. [1], our research directly targets improving performance on ML tasks. We specifically consider the whole ML pipeline and the exploration of task-specific models, which are not central to their works. Furthermore, their trial-and-error mechanism primarily serves debugging and paper writing, whereas ours is explicitly designed for performance optimization.
>
> In this regard, our MLE-STAR framework introduces two key novelties. First of all, orchestration of diverse agents is central to our multi-agent system's success. MLE-STAR uniquely integrates agents with distinct roles, such as the retriever, ablation study, planner, ensemble, and debugging agents. The seamless and effective connection of these specialized agents within our novel framework represents a significant and non-trivial advancement in multi-agent system deployment.
>
> Secondly, our targeted refinement strategy is an important innovation. Unlike previous machine learning agents that refined entire code blocks—limiting exploration of refinement strategies—we have introduced an ablation study agent. This novel component automatically analyzes the effectiveness of individual components within the ML pipeline, enabling exploration of diverse refinement strategies for a certain code block, leading to state-of-the-art performance in MLE tasks.
>
> We will incorporate this discussion and highlight the difference with Lu et al. [1] in the revised manuscript.
>
> [1] Lu et al., The AI Scientist: Towards Fully Automated Open-ended Scientific Discovery, arXiv 2024.
>
> ---
> **[Q1] The agent identifies one block and refines it at a time. Are there cases where multi-block refinement or full model resets (e.g., changing the model entirely) would be more effective?**
>
> Thanks for the valuable suggestions. Rather than identifying just one block and refining it at a time, selecting one or multiple blocks dynamically at a certain step seems to be an interesting future direction. This can be also implemented through the ablation study agent to select one or more code blocks to refine.
>
> On the other hand, changing the full code update might not be beneficial. In this rebuttal, we have conducted an additional ablation study to further validate the advantage of targeted refinement. Specifically, we removed the ablation study agent, allowing the agent to refine the entire script at every refinement step.
>
> As the below table illustrates, refining specific code blocks consistently outperforms refining the entire script. We hypothesize this is because focusing on extracted code blocks allows our agent to explore more diverse strategies for selected ML pipelines. We will incorporate these results and discussion in the revised manuscript.
> \begin{array}{lccccc}
> \hline
> \text{Method}&\text{Targeted refinement}&\text{DDD (RMSE)}&\text{DBI (Log Loss)}&\text{SAI (Log Loss)}&\text{WCR (AUROC)}\newline
> \hline
> \textbf{MLE-STAR (Variant)}&\text{X}&0.10818&0.45689&0.29141&0.98532\newline
> \textbf{MLE-STAR}&\text{O}&\textbf{0.06805}&\textbf{0.45351}&\textbf{0.27967}&\textbf{0.99028}\newline
> \hline
> \end{array}
>
> **Experimental settings:** We used the same four datasets – denoising-dirty-documents (DDD), dog-breed-identification (DBI), spooky-author-identification (SAI), and the-icml-2013-whale-challenge-right-whale-redux (WCR) – used in Table 3 of our manuscript. The experiments were repeated three times for each dataset and reported the average metrics (lower the better except WCR).
>
> ---
> **[Q2] Does the paper consider metrics beyond model performance, such as code readability and modularity?**
>
> We appreciate your valuable suggestions. Currently, our assessment focuses solely on model performance. However, we agree that code readability and modularity are also meaningful metrics. We plan to incorporate them into the revised manuscript as follows: (i) introducing them as additional optimization objectives alongside model performance, and (ii) adding a refinement stage that utilizes an LLM to enhance the readability of our final solution code.

---

> ### Comment · Reviewer_3VFF · 2025-08-01
> **Thanks for the responses**
>
> Thank the authors for the detailed responses. The additional results and the discussions are very helpful. I also agree that Fig. 8 in the paper already addresses some of my concerns, and appreciate the additional analysis. I have raised the score.

---

> > ### Author Response · Authors · 2025-08-05
> > **Thank you very much for the response**
> >
> > Dear reviewer 3VFF,
> >
> > Thank you very much for letting us know! We are happy to hear that our rebuttal addressed your questions well.\
> > If you have any further questions or suggestions, please do not hesitate to let us know.
> >
> > Thank you very much,\
> > Authors

---

### Official Review · Reviewer_RCp1 · 2025-06-26

**Clarity:** 3
**Significance:** 3
**Originality:** 2
**Rating:** 5
**Confidence:** 5

**Summary:**

This paper investigates the automated machine learning engineering with LLM agents. Specifically, this paper proposes MLE-STAR, an agent framework that leverages domain knowledge from the web search to develop SOTA machine learning models. Extensive experiments on MLE-Bench-Lite are provided to demonstrate the effectiveness of the proposed agent framework. MLE-STAR achieves medals in 44% of the Kaggle competitions from MLE-Bench-Lite, which is a notable progress.

**Questions:**

Will the authors open-source the MLE-STAR project? This is an important factor that may impact my recommendation. As this paper focuses on application instead of making technical contributions, if the authors would not open-source their project, I think this paper is less impactful for the community. If the authors can promise the open-sourcing, I would consider raising my score as accept.

**Ethical Concerns:**

["NO or VERY MINOR ethics concerns only"]

**Final Justification:**

I think this paper would be a strong submission for NeurIPS conditioned on the promised open-sourced project. Thus, I decide to increase my score to 5.

**Limitations:**

yes

**Quality:**

3

**Strengths And Weaknesses:**

Strengths:

1. The presentation quality of this paper is good and easy to follow for me.
2. The investigated problem is novel and interesting. Automated machine learning engineering with LLM agents may benefit the scientists in their scientific discoveries.
3. Experiments on ML-Bench-Lite are conducted to verify the effectiveness of the proposed agent framework.
4. MLE-STAR achieves medals in 44% of the tasks, which is a notable progress.

Weaknesses:

1. Please make rigorous claims in your paper. For example, in the abstract, the authors claim that

> Our experimental results show that MLE-STAR achieves medals in 44% of the Kaggle competitions on the MLE-bench, significantly outperforming the best alternative.

The experiments are conducted on **MLE-Bench-Lite**, instead of the complete version of MLE-Bench. Please revise this claim.

2. How is the web search enabled in MLE-STAR is not detailed in the paper. Do the authors simply utilize the DeepResearch function from Gemini to perform this? Or do the authors build an automatic workflow on their owns? Please provide more details on this.

3. As this paper focuses on novel applications of LLMs, please consider adding a new experiment to see whether MLE-STAR can achieve medals in real-time Kaggle competitions, instead of offline ones provided by MLE-Bench.

4. Why the authors utilize LLM as judge to analyze the novelty of the generated code script? MLE-Bench provides a tool that can identifies potential plagiarism. Please consider use that tool to make in-depth analyses.

---

> ### Author Rebuttal · Authors · 2025-07-30
>
> Dear Reviewer RCp1,
>
> We deeply appreciate your invaluable time and efforts in reviewing our manuscript. We respond to each of your comments one-by-one in what follows.
>
> ---
> **[W1] Please make rigorous claims in your paper (e.g., MLE-bench to MLE-bench Lite in abstract).**
>
> Thanks for your suggestion. We will clarify 'MLE-bench' to 'MLE-bench Lite' in the revised manuscript. For your interest, we are currently conducting experiments for the entire competition, and we will update the results once the experiments are complete.
>
> ---
> **[W2] Unclear details on web search implementation: How is web search enabled in MLE-STAR? Is DeepResearch from Gemini utilized, or is there an automatic workflow built by the authors? More details are needed.**
>
> We would like to clarify that MLE-STAR enables web search by connecting Google Search as a tool to the Gemini model (tool is named as grounding with google search).
>
> ---
> **[W3] Please conduct an experiment to evaluate MLE-STAR's performance in real-time Kaggle competitions, distinct from the offline evaluations provided by MLE-Bench, given the paper's focus on novel LLM applications.**
>
> We appreciate your constructive feedback. We would like to first clarify that our primary evaluation relies on the offline metrics provided by MLE-bench, since it allows for rigorous comparison with existing baselines. Also, these finished competitions provide a more robust and reliable assessment of medal achievements, because their leaderboards are finalized (note that the better solutions are often updated at the end of the competition). Nevertheless, we agree that evaluating our method in real-time Kaggle competitions is a compelling idea, and we will include such results in the revised manuscript.
>
> ---
> **[W4] Why do the authors utilize LLM as a judge to analyze the novelty of the generated code script? MLE-Bench provides a tool that can identify potential plagiarism. Please consider using that tool to make in-depth analyses. Please consider using a tool that can identify potential plagiarism, which is provided by MLE-bench.**
>
> Following your suggestion, we utilized Dolos, a source code plagiarism detection tool, to analyze submissions against the top associated notebooks from each Kaggle competition, as proposed by MLE-bench’s authors. Our analysis, summarized in the below table, shows that no submission-notebook pair exceeded a 60% similarity score (i.e., a criteria suggested from Appendix A.4 of the MLE-bench paper), indicating no detected instances of plagiarism.
> \begin{array}{lccc}
> \hline
> \text{Competition ID}&\text{Submission \\#1}&\text{Submission \\#2}&\text{Submission \\#3}\newline
> \hline
> \text{aerial-cactus-identification}&0.35&0.42&0.39\newline
> \text{aptos2019-blindness-detection}&0.27&0.29&0.27\newline
> \text{dog-breed-identification}&0.32&0.30&0.32\newline
> \text{dogs-vs-cats-redux-kernels-edition}&0.33&0.27&0.31\newline
> \text{histopathologic-cancer-detection}&0.32&0.36&0.35\newline
> \text{leaf-classification}&0.27&0.31&0.30\newline
> \text{plant-pathology-2020-fgvc7}&0.31&0.35&0.39\newline
> \text{ranzcr-clip-catheter-line-classification}&0.30&0.33&0.34\newline
> \text{siim-isic-melanoma-classification}&0.24&0.27&0.24\newline
> \text{denoising-dirty-documents}&0.25&0.27&0.27\newline
> \text{detecting-insults-in-social-commentary}&0.34&0.16&0.18\newline
> \text{jigsaw-toxic-comment-classification-challenge}&0.30&0.27&0.30\newline
> \text{random-acts-of-pizza}&0.37&0.39&0.37\newline
> \text{spooky-author-identification}&0.20&0.21&0.28\newline
> \text{new-york-city-taxi-fare-prediction}&0.35&0.33&0.34\newline
> \text{nomad2018-predict-transparent-conductors}&0.24&0.27&0.23\newline
> \text{tabular-playground-series-dec-2021}&0.26&0.19&0.29\newline
> \text{tabular-playground-series-may-2022}&0.24&0.32&0.20\newline
> \text{text-normalization-challenge-english-language}&0.22&0.19&0.20\newline
> \text{text-normalization-challenge-russian-language}&0.18&0.23&0.22\newline
> \hline
> \end{array}
>
> Note that the results of the mlsp-2013-birds competitions have been omitted because MLE-STAR was unable to generate submissions using Gemini-2.0-Flash. Additionally, the results of the icml-2013-whale-challenge-right-whale-redux competition have also been omitted because the authors of MLE-bench did not provide the relevant notebooks.
>
> ---
> **[Q1] Will the authors open-source the MLE-STAR project?**
>
> Yes, we will definitely open-source the MLE-STAR project. For reference, we wanted to provide the code as an anonymous link, but this was not possible due to rebuttal policy restrictions. We promise to make the code public.

---

> > ### Comment · Reviewer_RCp1 · 2025-08-01
> >
> > Thanks for the detailed responses. I think this paper would be a strong submission for NeurIPS conditioned on the promised open-sourced project. Thus, I decide to increase my score.

---

> > > ### Author Response · Authors · 2025-08-05
> > > **Thank you very much for the response**
> > >
> > > Dear reviewer RCp1,
> > >
> > > Thank you very much for letting us know! We are happy to hear that our rebuttal addressed your questions well.\
> > > If you have any further questions or suggestions, please do not hesitate to let us know.
> > >
> > > Thank you very much,\
> > > Authors

---

### Official Review · Reviewer_PksL · 2025-07-03

**Clarity:** 3
**Significance:** 3
**Originality:** 3
**Rating:** 5
**Confidence:** 4

**Summary:**

MLE-STAR introduces a machine learning engineering agent that uses Large Language Models (LLMs) augmented by external web search and targeted refinement. The agent constructs initial solutions by retrieving effective state-of-the-art model implementations via web searches, then iteratively refines these solutions through focused ablation studies of ML pipeline components. Additionally, MLE-STAR proposes an LLM-driven ensemble strategy to merge multiple candidate solutions. Evaluations on the MLE-bench Lite (22 Kaggle competitions) show substantial improvements over existing state-of-the-art agents (e.g., AIDE), with a medal achievement rate of approximately 44%.

**Questions:**

1. Can the authors explicitly quantify the incremental advantage of external web search versus relying solely on internal LLM knowledge? This was claimed without any empirical evidence.
2. How is novelty ensured against inadvertently using existing Kaggle solutions? Could more rigorous novelty checks (e.g., code similarity metrics) be included?
4. Have comparisons against traditional AutoML frameworks (e.g., AutoGluon, TPOT) been conducted? Could these results be explicitly included in the main manuscript?
5. Can the authors run their baselines and quantify performance across the full MLE-bench, rather than just the Lite version of the benchmark? To avoid any overfitting to the MLE-bench lite argument speculation.

**Ethical Concerns:**

["NO or VERY MINOR ethics concerns only"]

**Final Justification:**

The authors’ rebuttal clearly addressed my main concerns, especially regarding the benefit of web search and targeted refinement, with new ablation results and sensitivity analyses. Expanded baselines and improved novelty checks also strengthened the work. While some concerns about generalizability and resource demands remain, the authors have promised further discussion. Overall, the significant improvements and clarifications outweigh the remaining minor limitations. I have increased my score and now recommend Accept.

**Limitations:**

Authors have adequately addressed key limitations regarding potential data leakage. Additional discussion on computational resource demands, sensitivity analyses, and generalizability to non-Kaggle scenarios could further strengthen the manuscript.

**Paper Formatting Concerns:**

None identified.

**Quality:**

3

**Strengths And Weaknesses:**

## Strengths

* Quality: Clearly defined, technically rigorous methodology with robust experimental validation.
* Clarity: Well-organized, clearly articulated, easy-to-follow manuscript.
* Significance: Demonstrates substantial performance gains on a well-established benchmark, strongly impacting AutoML and LLM-agent research.
* Originality: Novel combination of external web search and fine-grained iterative refinement within an LLM-driven AutoML framework.

## Weaknesses



* Quality:
  * "MLE-STAR first leverages external knowledge by using a search engine to retrieve effective models from the web, forming an initial solution." This claim is supported by clear methodology; however, explicit ablations quantifying the specific incremental benefit of this external retrieval method versus relying solely on the LLM’s internal knowledge are missing.
  * "Iteratively refines it by exploring various strategies targeting specific ML components." Supported by detailed ablation studies and iterative refinement steps clearly outlined and empirically evaluated. However, specific evidence demonstrating the advantage of targeted refinement compared to whole-script revision is lacking.
* Significance: Potentially limited broader applicability beyond structured Kaggle-like competitions; unclear generalizability to more open-ended ML tasks.
* Originality:
  * Incremental rather than groundbreaking novelty, primarily integrating existing techniques.
  * Motivations for specific methodological choices, such as fixed numbers of iterations (outer/inner loops) and ensemble candidate counts, are insufficiently justified. Additional sensitivity analyses on these hyperparameters would significantly strengthen the methodological rigor.

### Missing References
* Zheng, L. et al. “Judging LLM-as-a-Judge with MT-Bench and Chatbot Arena.” NeurIPS, 2023.
* Lu, C. et al. “The AI Scientist: Towards Fully Automated Open-Ended Scientific Discovery.” arXiv 2024.
* Bersenev, D. et al. “Replicating a High-Impact Scientific Publication Using Systems of Large Language Models.” bioRxiv 2024.

---

> ### Author Rebuttal · Authors · 2025-07-30
>
> Dear Reviewer PksL,
>
> We deeply appreciate your invaluable time and efforts in reviewing our manuscript. We respond to each of your comments one-by-one in what follows.
>
> ---
> **[W1, W2, Q1] Ablation study on using search as a tool and the advantage of targeted refinement.**
>
> First, we have already shown some qualitative results in Figure 4 of the current manuscript that AIDE, which relies on internal LLM knowledge, often utilizes less competitive models compared to MLE-STAR. To further address your comments, we removed the search tool from Gemini-2.0-Flash (Variant 2) and ran the experiments with the same configuration as in the current manuscript. As shown in the table below, using a search as a tool consistently enhances the performance.
>
> In addition, we further validate the advantage of targeted refinement. Specifically, we removed the ablation study agent (Variant 1), allowing the agent to refine the entire script at every refining step.
>
> As the below table illustrates, refining specific code blocks consistently outperforms refining the entire script. We hypothesize this is because focusing on extracted code blocks allows our agent to explore more diverse strategies for selected ML pipelines.
>
> We will incorporate these results and discussion in the revised manuscript.
> \begin{array}{lcccccc}
> \hline
> \text{Method}&\text{Targeted refinement}&\text{Search as a tool}&\text{DDD (RMSE)}&\text{DBI (Log Loss)}&\text{SAI (Log Loss)}&\text{WCR (AUROC)}\newline
> \hline
> \textbf{MLE-STAR (Variant 1)}&\text{X}&\text{O}&0.10818&0.45689&0.29141&0.98532\newline
> \textbf{MLE-STAR (Variant 2)}&\text{O}&\text{X}&0.09303&0.65242&0.30529&0.96509\newline
> \textbf{MLE-STAR}&\text{O}&\text{O}&\textbf{0.06805}&\textbf{0.45351}&\textbf{0.27967}&\textbf{0.99028}\newline
> \hline
> \end{array}
>
> **Experimental settings:** We used the same four datasets – denoising-dirty-documents (DDD), dog-breed-identification (DBI), spooky-author-identification (SAI), and the-icml-2013-whale-challenge-right-whale-redux (WCR) – used in Table 3 of our manuscript. The experiments were repeated three times for each dataset and reported the average metrics (lower the better except WCR).
>
> ---
> **[W3] Unclear generalizability to more open-ended ML tasks.**
>
> We believe our method's applicability extends beyond Kaggle-like competitions, since it is designed to tackle any ML task, including open-ended problems, as long as metrics are given as natural language and data is available. Our agent can automatically and rigorously define metrics, and the data doesn't require extensive pre-cleaning.
>
> Furthermore, while MLE-bench is derived from Kaggle competitions, it encompasses a diverse array of tasks (e.g., audio classification, image denoising/classification, tabular classification/regressions). The effective performance of our method across these varied challenges underscores its broad applicability to real-world problems.
>
> ---
> **[W4] Incremental rather than groundbreaking novelty.**
>
> Our MLE-STAR framework introduces two critical and groundbreaking novelties not found in prior techniques.
>
> First of all, orchestration of diverse agents is central to multi-agent systems’ success. MLE-STAR uniquely integrates agents with distinct roles, such as the retriever, ablation study, planner, ensemble, and debugging agents. The seamless and effective connection of these specialized agents within our novel framework represents a significant and non-trivial advancement in multi-agent system deployment.
>
> Secondly, our targeted refinement strategy is another critical innovation. Unlike previous machine learning agents that refined entire code blocks—limiting exploration of refinement strategies—we have introduced an ablation study agent. This novel component automatically analyzes the effectiveness of individual components within the ML pipeline, enabling exploration of diverse refinement strategies for a certain code block, leading to state-of-the-art performance in MLE tasks.
>
> ---
> **[W5] Analyses on hyperparameters.**
>
> Following your suggestion, we have conducted a sensitivity analysis on the number of ensemble rounds. The below table indicates that while we utilized five rounds for ensemble strategy exploration, comparable performance can be achieved with fewer rounds. We will incorporate this detailed discussion in the revised manuscript. Furthermore, we commit to including additional analyses of other hyperparameters, such as the number of inner and outer loops.
> \begin{array}{ccccc}
> \hline
> \text{Round}&\text{DDD (RMSE)}&\text{DBI (Log Loss)}&\text{SAI (Log Loss)}&\text{WCR (AUROC)}\newline
> \hline
> 1&0.07147&\textbf{0.45351}&0.28164&0.98943\newline
> 3&\textbf{0.06805}&\textbf{0.45351}&\textbf{0.27967}&0.98898\newline
> 5&\textbf{0.06805}&\textbf{0.45351}&\textbf{0.27967}&\textbf{0.99028}\newline
> \hline
> \end{array}
>
> **Experimental settings:** Same as the above [W1, W2, Q1].
>
> ---
> **[W6] Missing references.**
>
> We would like to first note that the focuses of [1, 2, 3] differ significantly from our problem settings.
> - [1] primarily compares LLM-as-a-Judge and human preference alignment.
> - [2, 3] concentrate on automating the end-to-end scientific research process, from idea generation to experimental execution, using multi-agent frameworks.
>
> In contrast, our work focuses on machine learning engineering tasks. Our agent’s objective is to optimize solution performance given predefined task descriptions and datasets. Specifically, to achieve this, we propose new MLE-specific components, such as refining specific code blocks or ensembling solutions. On the other hand, [2] use a general approach aimed at writing good papers.
>
> While our core focus diverges, we do leverage LLM-as-a-Judge for specific sub-tasks, such as identifying data leakage (see Section 3.4 of the current manuscript) and assessing solution novelty (see Appendix H of the current manuscript). Furthermore, we acknowledge that [2, 3] utilize multi-agent systems like MLE-STAR. We will ensure proper citation and discussions of these relevant works in the revised manuscript.
>
> ---
> **[Q2] How is novelty ensured against existing Kaggle solutions? Could more rigorous novelty checks be included?**
>
> We would like to clarify that we have already performed novelty checks in Appendix H of the current manuscript. Specifically, using Gemini-2.5-Pro as a judge, we found that all the final solutions generated by our method are sufficiently novel compared to the top discussions of each competition.
>
> Moreover, to fully address your concern, we utilized Dolos, a source code plagiarism detection tool, to analyze submissions against the top associated notebooks from each Kaggle competition, as proposed by MLE-bench’s authors. Our analysis (please refer to [W4] of the Reviewer RCp1 due to the space limit) shows that no submission-notebook pair exceeded a 60% similarity score (i.e., a criteria suggested from Appendix A.4 of the MLE-bench paper), indicating no detected instances of plagiarism.
>
> ---
> **[Q3] Have comparisons against traditional AutoML frameworks been conducted? Could these results be explicitly included in the main manuscript?**
>
> First, we have already shown that MLE-STAR significantly outperforms AutoGluon in Table 9 of Appendix G in the current manuscript. Meanwhile, it is worth noting that such an AutoML framework is often restricted to data or task types (e.g., tabular data) and therefore cannot be a direct competitor. For example, AutoGluon should reformulate multi-class classification problems as multiple binary classification problems. Furthermore, such AutoML frameworks require continuous updates by domain experts to reflect the latest models, but our method automatically reflects these updates by utilizing search as a tool. We will explicitly include these results and discussions in the revised manuscript.
>
> ---
> **[Q4] Can the authors quantify performance across the full MLE-bench?**
>
> We appreciate the suggestion and are actively expanding our experimental evaluation across the full MLE-bench. Our intermediate results, using three seeds for each competition, are promising.
> - MLE-STAR has achieved three gold medals in both the stanford-covid-vaccine (SCV) and predict-volcanic-eruptions-ingv-oe (PVE) competitions. These are all high-complexity tasks (i.e., hard tasks), demonstrating our method's robust generalizability.
> - While MLE-STAR did not achieve a medal in cassava-leaf-disease-classification (CLD), its performance nearly reached the median threshold (0.89090).
>
> The table below presents a preliminary comparison with AIDE + o1-preview which consistently show MLE-STAR significantly outperforming AIDE + o1-preview. We are continuing additional experiments to update our results and will incorporate a comprehensive evaluation with full MLE-Bench in the revised manuscript.
> \begin{array}{lccccc}
> \hline
> \text{Competition ID}&\text{Task Complexity}&\text{Medals}&\text{Metric}&\text{AIDE + o1-preview}&\textbf{MLE-STAR + Gemini-2.5-Pro (Ours)}\newline
> \hline
> \text{CLD}&\text{Medium}&\text{No medal}&\text{Accuracy (\uparrow)}&0.86186&\textbf{0.88391}\newline
> \text{SCV}&\text{Hard}&\text{Gold}&\text{MCRMSE (\downarrow)}&0.40033&\textbf{0.23377}\newline
> \text{PVE}&\text{Hard}&\text{Gold}&\text{MAE (\downarrow)}&3747830&\textbf{2832981}\newline
> \hline
> \end{array}
>
> ---
> **[L1] Additional discussion on computational resource demands, sensitivity analyses, and generalizability to non-Kaggle scenarios could further strengthen the manuscript.**
>
> Thank you for your valuable feedback. We will enhance our discussion about limitations to include an analysis of the costs associated with utilizing commercial LLMs and integrate sensitivity analyses into the revised manuscript. Furthermore, we will expand on the applicability of MLE-STAR beyond Kaggle-like competitions, emphasizing that its ability to automatically define metrics and process uncleaned data makes it broadly relevant to any machine learning task.

---

> > ### Comment · Reviewer_PksL · 2025-08-06
> > **Response to Author Rebuttal**
> >
> > Dear Authors,
> >
> > Thank you for your detailed and thoughtful rebuttal. I appreciate the substantial effort invested in the new experiments, ablation studies, and clarifications. Your responses have directly addressed my primary concerns, particularly regarding the empirical benefit of web search and the targeted refinement strategy. The added results and analyses have strengthened my confidence in both the methodological rigor and the impact of your work, and I have updated my score accordingly.
> >
> > Looking ahead, I am curious whether your approach could be further enhanced by maintaining an open-ended, continually growing library of discovered code fragments or strategies—similar in spirit to the “Darwin Godel Machine” approach (Zhang et al., 2025)—to encourage continual improvement and knowledge reuse within the agent. Have you considered such mechanisms, or do you see opportunities for future work along these lines?
> >
> > Thank you again for your comprehensive and constructive rebuttal, and for the significant additional results and insights.

---

> > > ### Author Response · Authors · 2025-08-06
> > > **Thank you very much for the response**
> > >
> > > Dear Reviewer PksL,
> > >
> > > Thank you very much for letting us know! We are happy to hear that our rebuttal addressed your questions well.
> > >
> > > In addition, thank you for the valuable suggestion. Building a persistent, growing library of discovered strategies is a natural and exciting extension of our work. While our planner agent uses the scores of discovered strategies as feedback to make better suggestions, we also agree that finding better exploration strategies, such as tree search to refine targeted code block, is a promising direction for future work. We will incorporate such discussion on Darwin Godel Machine and future direction in the final draft.
> > >
> > > If you have any further questions or suggestions, please do not hesitate to let us know.
> > >
> > > Thank you very much,
> > >
> > > Authors

---

> ### Author Response · Authors · 2025-08-05
> **Further Discussion Before the Deadline**
>
> Dear Reviewer PksL,
>
> Thank you for your time and efforts again in reviewing our paper.
>
> We kindly remind that the discussion period will end soon (in a few days).
>
> We believe that we sincerely and successfully address your comments, with the results of the supporting experiments.
>
> If you have any further concerns or questions, please do not hesitate to let us know.
>
> Thank you very much!
>
> Authors

---

### Official Review · Reviewer_aeRQ · 2025-07-13

**Clarity:** 3
**Significance:** 2
**Originality:** 2
**Rating:** 4
**Confidence:** 4

**Summary:**

This paper introduces an agent framework, MLE-STAR, for orchestrating large language models to perform machine learning engineering tasks. This framework consists of a collection of sub-routines to perform specific tasks (e.g., search for an initial model and its implementation) and a protocol for invoking these sub-routines. It significantly outperforms baselines on MLE-Bench (a standard benchmark).

**Questions:**

Much of the methodology seems to be very focused on the MLE Bench task specifcially. Are there generalizable principles/takeaways here? Is this additional scaffolding something that will still be helpful as agents get better?

Since AIDE performance is reported for different models, why aren’t MLE-STAR results reported for different models (besides the Gemini-2.5-Pro ablation)?

How does the cost and running time of MLE-STAR compare to baselines like AIDE?

**Ethical Concerns:**

["NO or VERY MINOR ethics concerns only"]

**Final Justification:**

Thank you to the authors for response. I raised my score.

**Limitations:**

yes

**Quality:**

3

**Strengths And Weaknesses:**

Strengths:
+ This agent scaffolding introduced by this paper performs substantially better than prior methods.
+ The paper is for the most part clearly written and easy to follow.

Weaknesses:
- Lack of additional ablations: There seems to be a lot of room for insight from additional ablations here. For example, the authors claim that an LLM using its internal knowledge might be a limitation and use this to motivate searching for up-to-date models as an initial step. This seems like a very valuable ablation to run. I’d similarly like to see the benefits of each step in the proposed agent framework and the impact of choices such as the number of proposals at each step etc. I think that such additional analysis would significantly enrich this work and offer more broad takeaways.
- Unclear takeaways: This paper introduces an agent framework that is very specific to the MLE task. Many pieces of the framework like adding a data usage checker feel like manual patches for specific issues.

---

> ### Author Rebuttal · Authors · 2025-07-30
>
> Dear Reviewer aeRQ,
>
> We deeply appreciate your invaluable time and efforts in reviewing our manuscript. We respond to each of your comments one-by-one in what follows.
>
> ---
> **[W1] Lack of additional ablations (e.g., removing search at initialization, impact of choices such as the number of proposals at each step).**
>
> We appreciate your valuable feedback and have conducted additional experiments to further address your comments, building upon the ablation studies already presented (e.g., the effectiveness of our ensemble strategy in Table 4; performance improvements with advanced LLMs in Table 3).
>
> In this rebuttal, we've focused on two key components of the proposed approach:
> - **Targeted refinement strategy:** Our findings indicate that refining only the specific code block identified by the ablation study agent is more effective than refining the entire codebase.
> - **Integrating search as a tool:** Our experiments, conducted with the same configuration as outlined in the manuscript, consistently show that utilizing search as a tool significantly enhances the overall performance.
>
> These results, summarized in the table below, verify the effectiveness of our targeted refinement and the benefits of the search tool integration. We will incorporate these ablation studies in the revised manuscript.
> \begin{array}{lcccccc}
> \hline
> \text{Method}&\text{Targeted refinement}&\text{Search as a tool}&\text{DDD (RMSE)}&\text{DBI (Log Loss)}&\text{SAI (Log Loss)}&\text{WCR (AUROC)}\newline
> \hline
> \textbf{MLE-STAR (Variant 1)}&\text{X}&\text{O}&0.10818&0.45689&0.29141&0.98532\newline
> \textbf{MLE-STAR (Variant 2)}&\text{O}&\text{X}&0.09303&0.65242&0.30529&0.96509\newline
> \textbf{MLE-STAR}&\text{O}&\text{O}&\textbf{0.06805}&\textbf{0.45351}&\textbf{0.27967}&\textbf{0.99028}\newline
> \hline
> \end{array}
>
> Furthermore, we have conducted a sensitivity analysis on the number of ensemble rounds, and the below table indicates that while we utilized five rounds for ensemble strategy exploration, comparable performance can be achieved with fewer rounds. We will incorporate this detailed discussion in the revised manuscript. Also, we commit to including additional analyses of other hyperparameters, such as the number of inner and outer loops, in the revised manuscript.
> \begin{array}{ccccc}
> \hline
> \text{Round}&\text{DDD (RMSE)}&\text{DBI (Log Loss)}&\text{SAI (Log Loss)}&\text{WCR (AUROC)}\newline
> \hline
> 1&0.07147&\textbf{0.45351}&0.28164&0.98943\newline
> 3&\textbf{0.06805}&\textbf{0.45351}&\textbf{0.27967}&0.98898\newline
> 5&\textbf{0.06805}&\textbf{0.45351}&\textbf{0.27967}&\textbf{0.99028}\newline
> \hline
> \end{array}
>
> **Experimental settings:** We used the same four datasets – denoising-dirty-documents (DDD), dog-breed-identification (DBI), spooky-author-identification (SAI), and the-icml-2013-whale-challenge-right-whale-redux (WCR) – used in Table 3 of the current manuscript. The experiments were repeated three times for each dataset and reported the average metrics (lower the better except WCR).
>
> ---
> **[W2, Q1] Unclear takeaways: Very specific to the MLE task (e.g., data usage checker seems to be manual patches). Are there generalizable principles here? Is this additional scaffolding something that will still be helpful as the agent gets better?**
>
> Indeed, our method is a specialized multi-agent system for machine learning engineering (MLE) tasks. These tasks are incredibly diverse, encompassing areas like audio classification, image denoising/classification, and tabular regressions/classification. Therefore, effectively tackling MLE problems is crucial for addressing a wide range of real-world challenges. In this regard, we believe that our framework offers clear and significant advantages because it handles any MLE task where metrics are well-defined and data is readily available. Furthermore, our key ideas – leveraging search as a tool and refining targeted code blocks – have broader applicability to general coding agents. For example, a debugging agent could identify an erroneous code block and then focus on updating that specific section. We see these generalizable ideas as a promising future research direction.
>
> ---
> **[Q2] Why aren’t MLE-STAR results reported for different models?**
>
> First, we would like to share the overall results of the entire MLE-bench Lite using Gemini-2.5-Pro. As shown in the table below, Gemini-2.5-Pro shows significantly better performance than Gemini-2.0-Flash, recording a medal achievement rate of over 60%.
> \begin{array}{lccccc}
> \hline
> \text{Model}&\text{Above Median}&\text{Bronze}&\text{Silver}&\text{Gold}&\text{Any Medal}\newline
> \hline
> \text{Gemini-2.0-Flash}&63.6&\textbf{9.1}&4.5&30.3&43.9\newline
> \textbf{Gemini-2.5-Pro}&\textbf{83.3}&6.1&\textbf{21.2}&\textbf{36.4}&\textbf{63.6}\newline
> \hline
> \end{array}
>
> In addition, we conducted additional experiments using Claude-Sonnet-4 to further address your comments. As shown in the table below, other models besides Gemini also show promising results, proving compatibility and generalizability in terms of model types.
> \begin{array}{lcccc}
> \hline
> \text{Model}&\text{DDD (RMSE)}&\text{DBI (Log Loss)}&\text{SAI (Log Loss)}&\text{WCR (AUROC)}\newline
> \hline
> \text{Gemini-2.0-Flash}&0.06805&0.45351&0.27967&\textbf{0.99028}\newline
> \text{Claude-Sonnet-4}&\textbf{0.01547}&\textbf{0.31135}&\textbf{0.26101}&0.98878\newline
> \hline
> \end{array}
>
> **Experimental settings:** For the above table, we used the same four datasets – denoising-dirty-documents (DDD), dog-breed-identification (DBI), spooky-author-identification (SAI), and the-icml-2013-whale-challenge-right-whale-redux (WCR) – used in Table 3 of our manuscript. The experiments were repeated three times for each dataset and reported the average metrics (lower the better except WCR).
>
> ---
> **[Q3] How is the cost and running time of MLE-STAR compared to baselines like AIDE?**
>
> We would like to first clarify that MLE-STAR demonstrates comparable running times to the best alternative, AIDE. As detailed in Appendix F of the current manuscript, MLE-STAR averages 14.1 hours on MLE-bench Lite, while AIDE averages 15.4 hours on the same resource environment. This indicates our method does not require more execution time.
>
> We acknowledge that MLE-STAR requires higher cost due to increased token usage. As shown in the below table MLE-STAR requires 4.5 times more input and output tokens compared to AIDE. This is primarily because our method generates longer scripts during the targeted refinement and ensembling stages (i.e., the average line of the solution code for MLE-STAR in MLE-bench Lite is 414, while that for AIDE is 147). Still, with Gemini-2.0-Flash (cost is \\$0.15/0.60 per 1M input/output tokens), the cost of MLE-STAR is only about \$0.24 per each ML challenge.
>
> Despite the higher token cost, it is important to note that MLE-STAR consistently and significantly achieves better results than AIDE, as demonstrated throughout the current manuscript. We will incorporate this detailed analysis into the revised manuscript.
> \begin{array}{lcc}
> \hline
> \text{Method}&\text{\\# Input tokens}&\text{\\# Output tokens}\newline
> \hline
> \text{AIDE}&\text{194K}&\text{38K}\newline
> \text{MLE-STAR (Ours)}&\text{874K}&\text{175K}\newline
> \hline
> \end{array}

---

> ### Author Response · Authors · 2025-08-05
> **Further Discussion Before the Deadline**
>
> Dear Reviewer aeRQ,
>
> Thank you for your time and efforts again in reviewing our paper.
>
> We kindly remind that the discussion period will end soon (in a few days).
>
> We believe that we sincerely and successfully address your comments, with the results of the supporting experiments.
>
> If you have any further concerns or questions, please do not hesitate to let us know.
>
> Thank you very much!
>
> Authors

---

> > ### Comment · Area_Chair_XPFs · 2025-08-06
> > **Please engage**
> >
> > Dear reviewer aeRQ, please read the author response and let us know if they have addressed your concerns or not. Note that the discussion period ends Aug 8th AoE, but the earlier the better.
> >
> > Best,
> > Your AC

---

### Decision · Program_Chairs · 2025-09-17

**Decision:**

Accept (poster)

**Comment:**

The reviewers converged on a positive assessment of the paper, highlighting that MLE-STAR substantially outperforms prior LLM-based ML engineering agents on MLE-Bench Lite and introduces a compelling combination of external web search, targeted refinement via ablation, and ensemble strategies. Strengths cited include clear presentation, strong empirical performance, and a meaningful contribution to the growing field of agentic systems for ML engineering. The authors’ rebuttal further strengthened the case, providing new ablations demonstrating the benefit of search and targeted refinement, sensitivity analyses on hyperparameters, plagiarism checks with "Dolos", and clarifications on resource usage and implementation. These additions addressed most major reviewer concerns, leading multiple reviewers to raise their scores to Accept.

The main disagreements were around novelty and generalizability. Some reviewers felt that the contributions were incremental, highly tailored to MLE-Bench, or patched for specific issues. Others argued that the orchestration of specialized agents and the targeted refinement strategy represent significant and broadly useful innovations. Concerns about cost, resource demands, and generalizability beyond Kaggle-like tasks remain, but reviewers generally considered these outweighed by the empirical impact and strengthened methodological justification.